# Lactate, Heart Rate and Rating of Perceived Exertion Responses to Shorter and Longer Duration CrossFit^®^ Training Sessions

**DOI:** 10.3390/jfmk3040060

**Published:** 2018-11-28

**Authors:** Ramires Alsamir Tibana, Nuno Manuel Frade de Sousa, Jonato Prestes, Fabrício Azevedo Voltarelli

**Affiliations:** 1Graduate Program in Health Sciences, Faculty of Medicine, Federal University of Mato Grosso, 78000 Cuiabá (MT), Brazil; 2Laboratory of Exercise Physiology, Faculty Estacio of Vitoria, 29010 Vitoria (ES), Brazil; 3Graduate Program of Physical Education, Catholic University of Brasilia, 04534 Brasilia (DF), Brazil

**Keywords:** extreme conditioning training, metabolic stress, cardiovascular response, high intensity functional training

## Abstract

The aim of this study was to analyze blood lactate concentration (LAC), heart rate (HR), and rating perceived exertion (RPE) during and after shorter and longer duration CrossFit^®^ sessions. Nine men (27.7 ± 3.2 years; 11.3 ± 4.6% body fat percentage and training experience: 41.1 ± 19.6 months) randomly performed two CrossFit^®^ sessions (shorter: ~4 min and longer: 17 min) with a 7-day interval between them. The response of LAC and HR were measured pre, during, immediately after, and 10, 20, and 30 min after the sessions. RPE was measured pre and immediately after sessions. Lactate levels were higher during the recovery of the shorter session as compared with the longer session (shorter: 15.9 ± 2.2 mmol/L/min, longer: 12.6 ± 2.6 mmol/L/min; *p* = 0.019). There were no significant differences between protocols on HR during (shorter: 176 ± 6 bpm or 91 ± 4% HRmax, longer: 174 ± 3 bpm or 90 ± 3% HRmax, *p* = 0.387). The LAC was significantly higher throughout the recovery period for both training sessions as compared to pre-exercise. The RPE was increased immediately after both sessions as compared to pre-exercise, while there was no significant difference between them (shorter: 8.7 ± 0.9, longer: 9.6 ± 0.5; *p* = 0.360). These results demonstrated that both shorter and longer sessions induced elevated cardiovascular responses which met the recommendations for gains in cardiovascular fitness. In addition, both training sessions had a high metabolic and perceptual response, which may not be suitable if performed on consecutive days.

## 1. Introduction

The regular practice of exercise is associated with the reduction of morbidity and mortality [1]. Nevertheless, epidemiological data reveal that the majority of adult population do not meet the recommended levels of physical activity [2]. Sedentarism contributes to the global epidemic of cardiovascular diseases, obesity, diabetes, hypertension, and dyslipidemias [3]. One of the main issues for the failure in participating of regular exercise is the lack of time or perceived lack of time [4]. Thus, the development of exercise interventions intended to counteract chronic diseases and overwhelm the perceived lack of time has an important public health value.

Alternative training modalities have gained popularity in recent years. According to the American College of Sports Medicine (ACSM), high intensity interval training (HIIT), training with body weight, resistance training, and group training are the main fitness trends of 2019 [5], showing that there was an increase in research of high intensity alternative training modalities. However, concern has been raised about the elevated metabolic and cardiovascular stress, accompanied by the possibility of higher rates of non-functional overreaching [6,7], injuries [7,8], and rhabdomyolysis [7,9].

An alternative to traditional HIIT is the CrossFit^®^ that consists of performing functional and sports exercises. This type of training utilizes Olympic weightlifting exercises, such as snatch and clean, basic exercises including squats, deadlifts, and bench press, rowing, running and cycle ergometer cyclic exercises, and gymnastic movements, such as parallels, rings, and bars [7]. These exercises are performed as fast as possible, with an elevated number of repetitions, and a limited rest interval in a circuit format to develop general conditioning, commonly defined as metabolic conditioning in CrossFit^®^ modality. High intensity training also has been associated with a more pronounced participation of the general population as compared with moderate aerobic training and resistance training [10].

Although some research has evaluated physiological responses to CrossFit^®^ training [6,11,12,13], to date, only the study from Kliszczewicz, et al. [14] has compared heart rate (HR) response during a CrossFit^®^ training (Cindy: A benchmark workout that consist of as much sets as possible within 20 min: five pull-ups, 10 push-ups and 15 air-squats) with high intensity treadmill training (20 min at 90% of HRmax). The mean HR for 20 min was above 93% of HRmax, and significantly higher in the high intensity treadmill training. This HR response meets the intensity recommendations from ACSM to improve cardiorespiratory fitness [15] and previous research reported modest improvements in aerobic fitness after some weeks of high-intensity functional training [16,17]. Although CrossFit^®^ includes a dynamic of movements with elevated repetitions and high intensity, training protocols have distinct duration, from shorter protocols of only some minutes (2 to 8 min) to longer sessions (20 to 30 min), which may directly affect cardiovascular, metabolic, and rating of perceived exertion responses. We are not aware of any study evaluating these responses during shorter versus longer CrossFit sessions.

Thus, the aim of the present study was to compare blood lactate (LAC), HR and rating of perceived exertion (RPE) responses during and after shorter versus longer duration CrossFit^®^ sessions. The hypothesis is that the cardiovascular response is higher than the recommended values by the ACSM with a greater metabolic and RPE response for both protocols.

## 2. Materials and Methods

### 2.1. Subjects

Nine apparently health men participated in the present study (27.7 ± 3.2 years; 11.3 ± 4.6% body fat percentage). The inclusion criteria were as follows: No muscle, joint, or bone injuries, presence of any disease that could compromise health during the study; negative response to the Physical Activity Readiness Questionnaire; signed the written informed consent document; have a minimal experience of six months with CrossFit^®^ practice. The study was approved by the Research Ethics Committee from the local University (Protocol number: 2.698.225; 7 June 2018).

### 2.2. Experimental Design 

Subjects visited the laboratory on two occasions with 7 to 9 days apart. They completed a shorter (Fran), and a longer CrossFit^®^ session (Fight Gone Bad). During the tests, HR was measured, while before and after the protocols, RPE and LAC were evaluated. Subjects were instructed to avoid meals two hours before sessions, to avoid alcoholic drinks, and stimulating drinks (containing caffeine, ephedrine, etc.) in the 2 days preceding the experimental session. Medication or supplementation that could change LAC and HR were also suspended for 2 days preceding the experimental session. When medication could not be suspended, the subjects were excluded. The subjects were counterbalanced in regards to the order of the training sessions; the sessions were randomly chosen and completed at the same time of day in the presence of the same researchers under controlled room temperature. The subjects were adapted to both CrossFit^®^ sessions and they recognized Fran and FGB sessions as described elsewhere [18], as they had six months of previous experience in the modality.

### 2.3. CrossFit^®^ Sessions

The shorter duration CrossFit^®^ session was Fran, which consists of 21-15-9 repetitions of thrusters (43 kg) and pull-ups as fast as possible (Figure 1). The dynamics of the sessions is to complete 21 thrusters followed by 21 pull-ups, 15 thrusters and 15 pull-ups, 9 thrusters and 9 pull-ups. The time to complete all repetitions was monitored. Subjects could rest at any during sessions, restarting from the last repetition completed.

The longer duration CrossFit^®^ session was Fight gone bad (FGB), which consists of three sets of 5 min per set, during which, as many repetitions as possible in five exercises are attempted (wall ball (9 kg), sumo deadlift high-pull (34 kg), box jump (50 cm), push-press (34 kg), and rowing (calories), respectively; Figure 1). Exercises are organized in stations, while the subject completes 1 min for each exercise with a 1 min rest interval between sets. The clock does not restart or stop between exercises. The result of the session is the sum of repetitions completed in each exercise for the three sets.

### 2.4. Blood Lactate Concentration (LAC)

Before training protocols, subjects remained seated for 10 min to collect blood samples for LAC measures by a portable device (Roche, Accutrend Plus system, São Paulo, Brazil) [19]. Lactate was also measured immediately after, 10, 20, and 30 min after both training sessions. Gases and 70% alcohol were used for asepsis of the finger to puncture the lateral pulp by using a lancet, and disposable gloves. A blood drop was inserted in the center of the test zone of the reactive tape to measure LAC. 

### 2.5. Heart Rate Analysis (HR)

The monitoring of HR was done by Polar H10 HR-monitor (Polar Electro Oy, Kemple, Finland), with recording intervals of 1 s. All subjects were monitored during both sessions: Shorter duration varying from 2 min 20 s to 9 min 01 s; and longer duration with a fixed time of 17 min (15 min of exercise, and 2 min of rest). Moreover, after the protocols, subjects remained seated for 30 min for the analysis of HR as follows: Immediately after, 10, 20-, and 30-minutes following exercise. Maximal heart rate was estimated using the age-predicted equation (220-age).

### 2.6. Rating of Perceived Exertion (RPE)

The RPE was measured before and immediately after exercise by the RPE CR10 Borg scale adapted by Foster et al. [20], as instrument composed by a Likert type scale of 11 points, varying from 0 to 10, initiated with “very, very light“ and terminated with “very, very hard “. The anchorage of the RPE scale was conducted in the beginning of both protocols by memory anchorage to standardize low and high RPE. Information about the RPE was explained to the subjects individually according to the recommendations from Foster et al. [20].

Briefly, the following information was provided: “The perceived exertion is defined the effort intensity, stress, discomfort, and fatigue felt during exercise. Utilize the numbers of this scale to report how your body feels during exercise. The number zero in the scale describes “minimal effort” and represents your lowest imaginable effort. The number 10 described “maximum effort” and represents the highest imaginable effort. If you feel an exertion between extremely easy and maximum effort indicate a number between 0 and 10. At the end the tests we will require you to point a number to inform us about how your body is feeling in general. There are no right or wrong numbers. The verbal descriptors may help you to choose a number” [20].

During the anchoring procedure, a RPE scale was presented to the subject by another evaluator that was present in the evaluation room. Subjects also received a copy of the scale with the respective instructions for anchorage. This was provided for subjects to read during the general warm-up for each session [20].

### 2.7. Statistical Analysis

The data are expressed as mean value ± standard deviation (SD). Shapiro–Wilk test was applied to check for normal distribution of study variables. In case of non-normal distribution, logarithmic transformation was performed. A two-way repeated measures ANOVA was used to compare the LAC, HR, and RPE between short and long sessions of CrossFit^®^. Compound sphericity was verified by the Mauchley test. When the assumption of sphericity was not met, the significance of F-ratios was adjusted according to the Greenhouse–Geisser procedure. Tukey’s post-hoc test with Bonferroni adjustment was applied in the event of significance. Paired sample t test was used to compare HR during exercise between CrossFit^®^ sessions. The achieved power of the sample size was calculated based on the interaction LAC between short and long sessions of CrossFit^®^. The effect size *f* was 0.302 and the achieved power was 0.801. The level of significance was *p* ≤ 0.05 and SPSS version 20.0 (Somers, NY, USA) software was used.

## 3. Results

The average time for Fran was 4.11 ± 2.36 min, the fastest time was 2.20 min and the slowest time was 9.01 min. The FGB was a fixed time of 17 min and the average number of repetitions were 294.1 ± 32.3 repetitions.

There was a statistically significant two-way interaction between CrossFit^®^ sessions and recovery time on LAC (*p* = 0.018; Figure 2). LAC was significantly higher throughout recovery time than rest for both CrossFit^®^ sessions (Fran: 2.1 ± 0.4 mmol/L at rest, 17.7 ± 4.2 mmol/L immediately after session, 17.2 ± 2.2 mmol/L after 10 min, 15.7 ± 2.3 after 20 min, and 11.9 ± 1.7 after 30 min; FGB: 2.0 ± 0.7 mmol/L at rest, 16.2 ± 2.9 mmol/L immediately after session, 14.4 ± 3.5 mmol/L after 10 min, 11.1 ± 3.6 after 20 min, and 8.7 ± 2.8 after 30 min. However, LAC during recovery time of Fran was significantly higher (*p* = 0.016) than the recovery time of FGB. The 30 min area under the curve of LAC after Fran was also significantly higher than the area under the curve of LAC after FGB (Fran: 15.9 ± 2.2 mmol/L/min; FGB: 12.6 ± 2.6 mmol/L/min; *p* = 0.019).

Figure 3 shows the HR during the CrossFit^®^ sessions. When the average and peak HR during sessions were compared, there was no statistically significant difference between CrossFit^®^ sessions (Fran: 176 ± 6 bpm or 91 ± 4%HRmax average; FGB: 174 ± 3 bpm or 90 ± 3%HRmax average; *p* = 0.387; Fran: 186 ± 5 bpm peak; FGB: 185 ± 3 bpm peak). There was also no statistically significant difference between the average HR during Fran and average HR during the first set of the FGB (170 ± 5 bpm or 88 ± 3%HRmax; *p* = 0.060), second set (175 ± 3 bpm or 91 ± 3%HRmax; *p* = 0.886), and third set (179 ± 4 bpm or 93 ± 3%HRmax; *p* = 0.168). During Fran CrossFit^®^ session, 90% of HRmax was achieved 30 s after the start of the session, on the other hand, 90 s was required to achieve 90% of HRmax during FGB CrossFit^®^ session.

There was no two-way interaction between CrossFit^®^ sessions and recovery time on HR (*p* = 0.399; Figure 4). HR throughout the recovery time was significantly higher than the rest time for the two CrossFit^®^ sessions, however, without statistically significant differences between sessions.

There was also no two-way interaction between CrossFit^®^ sessions and time on RPE (*p* = 0.360; Figure 5).

## 4. Discussion

The present study compared LAC, HR, and RPE during and after shorter versus longer CrossFit^®^ sessions. The main results demonstrated that: (i) The shorter duration protocol displayed a higher LAC response during the recovery period as compared with the longer duration; (ii) both protocols reached a mean superior to 90% HRmax during training, without differences between them, and (iii) the RPE was close to the maximum for both protocols, without differences between them.

To the best of our knowledge, this is the first study to compare different durations of CrossFit^®^ sessions (shorter vs. longer) on cardiovascular, metabolic, and RPE responses in trained men. Previous studies have reported these results, but only separately, and in response to different training models [6,11,12,13,14]. For example, Tibana et al. [6,12] analyzed metabolic, cardiovascular, and RPE responses during two protocols of similar duration (10 min vs. 12 min) in trained men and found differences between them in metabolic (LAC: 11.8 ± 1.3 vs. 9.05 ± 2.56 mmol/L), cardiovascular (86 ± 11% vs. 82 ± 12% HRmax), and RPE (8.8 ± 1.2 vs. 8.0 ± 1.2). It is noteworthy that the shorter duration protocol had a weightlifting exercise (snatch), while the longer protocols were composed only of cardiovascular exercises (rowing and burpees), which may have influenced the respective responses. Fernandez-Fernandez, et al. [13] were the first to evaluate Cindy and Fran CrossFit^®^ workouts. They showed that both workouts analyzed are high intensity workouts achieving near maximal physiological (e.g., 95% to 95% of HRmax) and perceptual levels (e.g., RPE values >8). Recently, Kliszczewicz, et al. [11] analyzed the cardiovascular and autonomic responses during and after two CrossFit^®^ sessions with different duration (< 5 min and 15 min). The results demonstrated no difference in cardiovascular response between sessions (92.7 ± 4% vs. 91.3 ± 3% HRmax). Similarly to Kliszczewicz, et al., [11] the results of the present study showed no differences in HRmax reached during exercise and HR measured after 30 min of both sessions, albeit the wide difference in duration. A novel finding of the present study was that RPE and LAC was significantly elevated immediately after sessions and higher LAC was recorded during the recovery period of the shorter duration sessions, indicating an exacerbated effect of exercise intensity. Although the cardiovascular stress in the longer duration protocol can be increasingly attributed to the accumulative effects of repetitions along time, the cardiovascular stress in the shorter duration session can be more attributed to intensity, which also resulted in higher LAC, that is, higher metabolic stress. This information is extremely relevant to determine the type of physiological stress that is applied to an individual during training and the planning that is required in regards to the logic programming of the training, without forgetting the issue of cardiovascular safety. Besides that, Perciavalle, et al. [21] also showed that athletes practicing CrossFit^®^, with high levels of blood lactate, should consequently have attentional performances somewhat limited. Thus, reaction time, execution time, number of errors, and number of omissions that exhibited a significant worsening concomitantly with the increase in blood lactate, could be changed during both short and long sessions.

An important finding of the present study is that CrossFit^®^ training sessions meet the intensity recommended by the ACSM to improve cardiovascular fitness [15], regardless of session duration. Interestingly, previous research reported modest improvements in aerobic fitness after some weeks of high-intensity functional training [16,17], despite the wide variety of exercises. On the other hand, both protocols elicited an elevated metabolic stress and RPE. Thus, practitioners must be advised to avoid performing daily sessions at the highest intensity possible to prevent possible non-functional overreaching and injuries. Considering the limited number of studies conducted, more evidence is required to evaluate different intensities of CrossFit^®^ sessions.

Despite the interesting findings of this study, some limitations need to be mentioned. First, the reduced number of subjects. Second, it should be noted that these results should be considered only for trained male subjects. Therefore, our findings may not be directly transferable to the untrained or to females.

## 5. Conclusions

Results revealed that both shorter and longer duration sessions elicited high cardiovascular responses, which meets the recommendations to improve cardiovascular fitness. The elevated metabolic and RPE responses observed in both protocols indicate that they may not be suitable on consecutive days without sufficient recovery.

## Figures and Tables

**Figure 1 jfmk-03-00060-f001:**
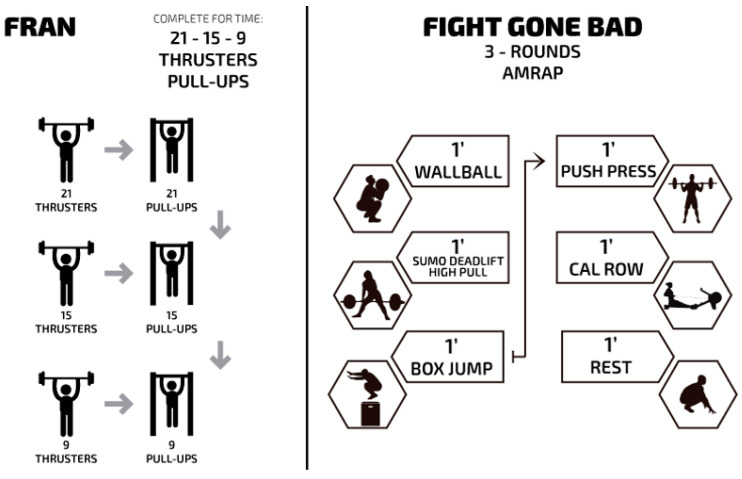
Description of training protocols.

**Figure 2 jfmk-03-00060-f002:**
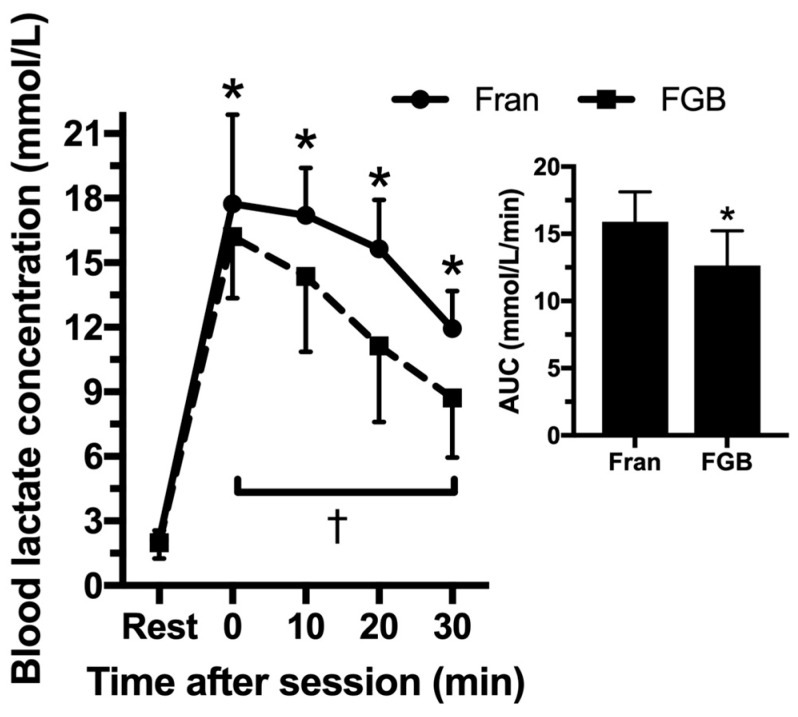
Blood lactate concentration before and after CrossFit^®^ sessions and area under the curve (AUC) of blood lactate concntration during the 30 min of recovery for both sessions. * *p* ≤ 0.05 for rest for the two CrossFit^®^ sessions; † *p* ≤ 0.05 between CrossFit^®^ sessions.

**Figure 3 jfmk-03-00060-f003:**
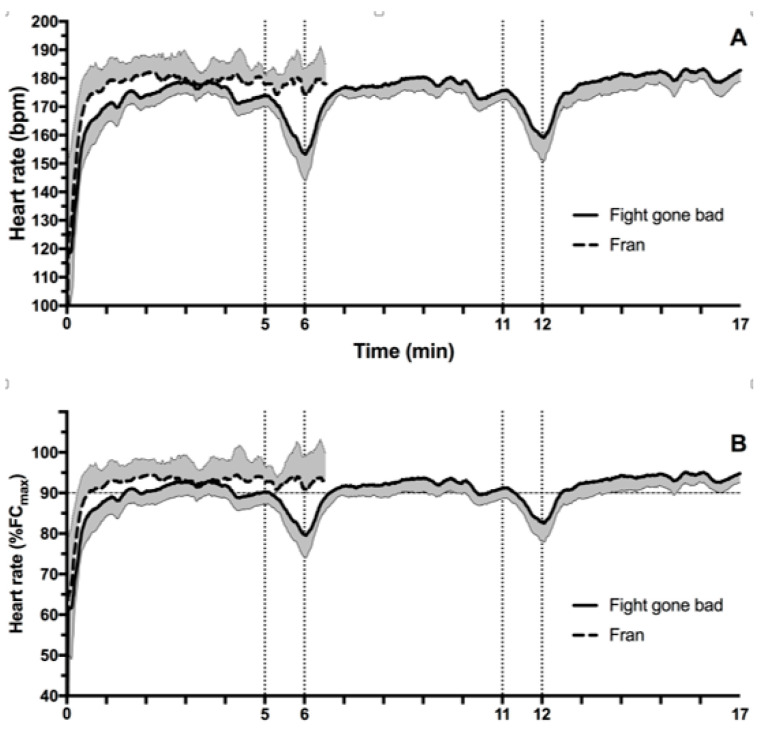
Heart rate (HR) during CrossFit^®^ sessions. (**A**) Absolute values; (**B**) relative values of HRmax using the 220-age equation. The shaded area corresponds to the standard deviation of HR during sessions.

**Figure 4 jfmk-03-00060-f004:**
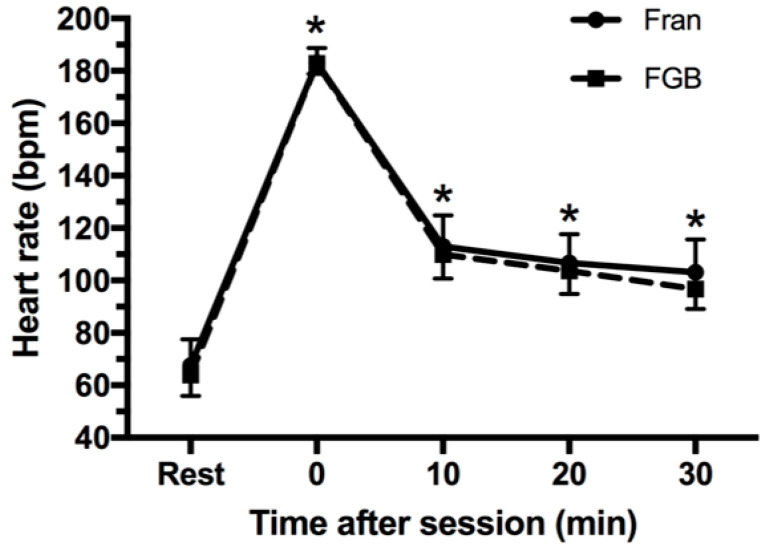
Heart rate (HR) before and after CrossFit^®^ sessions. * *p* ≤ 0.05 for rest for the two CrossFit^®^ sessions.

**Figure 5 jfmk-03-00060-f005:**
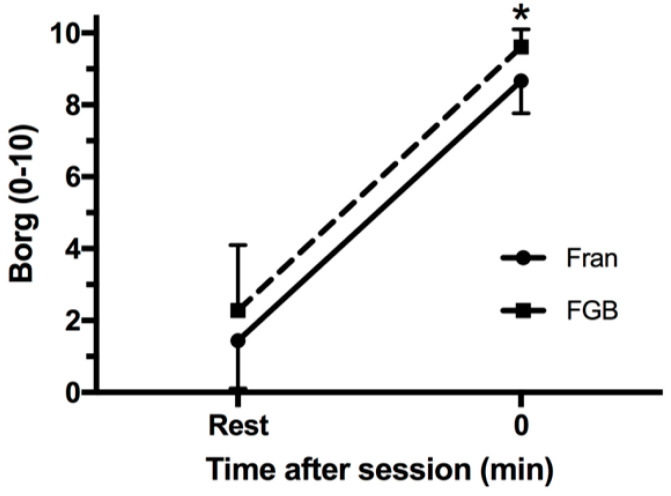
Rating of perceived exertion (RPE; 0 to 10) before and immediately after CrossFit^®^ sessions. * *p* ≤ 0.05 for rest for the two CrossFit^®^ sessions.

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
