# Peer review of "Lactate, Heart Rate and Rating of Perceived Exertion Responses to Shorter and Longer Duration CrossFit® Training Sessions"

_jfmk, 2018, doi:10.3390/jfmk3040060_

Round 1

Reviewer 1 Report

Here, two different setting of cross fit training are compared with respect to the hart rate, lactate and rating perceived exertion. In general it is an research question worth to study. There are some minor issues with the manuscript that should be addressed.

- l16. the most important characteristics is not the experience. Please round this number up. If it is important, include it into the method section. 

-l58. Who is Cindy? 

-chapter 2.3 maybe an image could explain the difference between Fran and fgb?

- For me it is not entirely clear that this should be a cardiovascular tines training. Please elaborate this point more in the discussion. 

Author Response

We appreciate the comments. A point-by-point response is below:

- l16. the most important characteristics is not the experience. Please round this number up. If it is important, include it into the method section. 

Response:  We agree with the reviewer. We added additional information. In this issue, the sentences “Nine men (27.7 ± 3.2 years; 11.3 ± 4.6 % body fat percentage and training experience: 41.1 ± 19.6 months) randomly performed two CrossFit® sessions (shorter: ~ 4 min and longer: 17 min) with 7-day interval between them.” was changed.

-l58. Who is Cindy? 

Response: Thanks for this comment. Cindy is a benchmark workout that consist of as much sets as possible within 20 minutes: 5 pull-ups, 10 push-ups and 15 air-squats. The sentence was changed.

-chapter 2.3 maybe an image could explain the difference between Fran and fgb?

Response: Thanks for this comment. We added an image (figure 1).

- For me it is not entirely clear that this should be a cardiovascular tines training. Please elaborate this point more in the discussion. 

Response:  Thanks for this comment. However, previous research reported modest improvements in aerobic fitness after some weeks of high-intensity functional training [Brisebois et al., 2018; Nieuwoudt et al., 2017] which demonstrates that this type of training can be considered cardiovascular training. The citations and a sentence were added in the discussion. 

Reviewer 2 Report

This manuscript sought to determine differences between a short and a long bout of Crossfit. It was noted that the HR responses were similar between conditions, lactate was elevated after the short duration CrossFit compared to the long, and RPE was elevated after both conditions, with no difference between them.  While this manuscript is well presented, there are some nuances that need to be addressed.

-The manuscript suffers from a lack of flow. The title is, 'HR, lactate, RPE', but the hypothesis is 'Lactate, HR, RPE'. The results are presented as 'HR, lactate and RPE', but the Discussion is 'lactate, HR, RPE.' This makes it really difficult in the reader, and makes it really hard to follow.

-Page 2, lines 80-87. Did the authors control for medications? Supplements?

-PAge 2, lines 80-87. Why are women are not included?  This is a severe limitation in my opinion.

-Page 2, lines 80-87. Why was the decision made to randomly assign the conditions, versus counter-balancing them? It seems that counterbalancing them would be more appropriate.

-Page 3, lines108-113. What was the ICC of your lactate monitor?

-Page 6, lines 207-209. I don't understand why there is mention of HRV (RMSSD, HF). they seem out of place.

Author Response

We appreciate the comments. A point-by-point response is below:

-The manuscript suffers from a lack of flow. The title is, 'HR, lactate, RPE', but the hypothesis is 'Lactate, HR, RPE'. The results are presented as 'HR, lactate and RPE', but the Discussion is 'lactate, HR, RPE.' This makes it really difficult in the reader, and makes it really hard to follow.

Response:  We agree with the reviewer. We changed the order.

-Page 2, lines 80-87. Did the authors control for medications? Supplements?

Response: Medication or supplementation that could change LAC and HR were also suspended for 2 days preceding the experimental session. When medication could not be suspended, the subjects were excluded. We added the sentence to the section 2.2.

-PAge 2, lines 80-87. Why are women are not included?  This is a severe limitation in my opinion.

Response:  We agree with the reviewer and we added additional information about this limitation.

In this issue, the sentences “Despite the interesting findings of this study, some limitations need to be mentioned. First, the reduced number of subjects. Second, it should be noted, that these results should be considered only for trained men subjects. Therefore, our findings may not be directly transferable to untrained or to females.”

-Page 2, lines 80-87. Why was the decision made to randomly assign the conditions, versus counter-balancing them? It seems that counterbalancing them would be more appropriate.

Response: We randomly assign the order of the training sessions, not the first training session for each subject. Is this way, the subjects are counterbalancing. We adjusted the sentence.

-Page 3, lines108-113. What was the ICC of your lactate monitor?

Response: Thanks for this comment. In our study we used the Accutrend Lactate which according to Baldari et al., 2009 have good accuracy and their high reliability and linearity.

Baldari C, Bonavolontà V, Emerenziani GP, Gallotta MC, Silva AJ, Guidetti L. Accuracy, reliability, linearity of Accutrend and Lactate Pro versus EBIO plus analyzer.  Eur J Appl Physiol. 2009 Sep;107(1):105-11.

-Page 6, lines 207-209. I don't understand why there is mention of HRV (RMSSD, HF). they seem out of place.

Response:  The result was fro m the study of Kliszczewicz, Williamson, Bechke, McKenzie and Hoffstetter [10]. The sentence was removed.